# Molecular Characterization and Identification of Potential Inhibitors for ‘E’ Protein of Dengue Virus

**DOI:** 10.3390/v14050940

**Published:** 2022-04-29

**Authors:** Rishi Gowtham Racherla, Sudheer Kumar Katari, Alladi Mohan, Umamaheswari Amineni, Manohar Badur, Abhijit Chaudhury, Mudhigeti Nagaraja, Sireesha Kodavala, Meenakshi Kante, Usha Kalawat

**Affiliations:** 1Department of Clinical Virology, Sri Venkateswara Institute of Medical Sciences (SVIMS), Tirupati 517507, Andhra Pradesh, India; rishigowtham@gmail.com (R.G.R.); nagarajamudhigeti@gmail.com (M.N.); kantemeenakshi@gmail.com (M.K.); 2Department of Bioinformatics, Sri Venkateswara Institute of Medical Sciences (SVIMS), Tirupati 517507, Andhra Pradesh, India; katarisudheerkumar@gmail.com (S.K.K.); amineni.maheswari@gmail.com (U.A.); 3Department of Medicine, Sri Venkateswara Institute of Medical Sciences (SVIMS), Tirupati 517507, Andhra Pradesh, India; alladimohan@rediffmail.com; 4Department of Pediatrics, Sri Venkateswara Ramnarain Ruia Government General Hospital (SVRRGGH), Tirupati 517507, Andhra Pradesh, India; punya_manohar2002@yahoo.com; 5Department of Microbiology, Sri Venkateswara Institute of Medical Sciences (SVIMS), Tirupati 517507, Andhra Pradesh, India; ach1964@rediffmail.com; 6Department of Biotechnology, Sri Venkateswara Institute of Medical Sciences (SVIMS), Tirupati 517507, Andhra Pradesh, India; sireeshabiotech140@gmail.com

**Keywords:** dengue virus, genotypes and serotypes, molecular dynamics simulation, envelope protein inhibitors

## Abstract

Dengue is an arthropod-borne acute febrile illness caused by Dengue Virus (DENV), a member of *Flaviviridae*. Severity of the infection ranges from mild self-limiting illness to severe life-threatening hemorrhagic fever (DHF) and dengue shock syndrome (DSS). To date, there is no specific antiviral therapy established to treat the infection. The current study reports the epidemiology of DENV infections and potential inhibitors of DENV ‘E’ protein. Among the various serotypes, DENV-2 serotype was observed more frequently, followed by DENV-4, DENV-1, and DENV-3. New variants of existing genotypes were observed in DENV-1, 2, and 4 serotypes. Predominantly, the severe form of dengue was attributable to DENV-2 infections, and the incidence was more common in males and pediatric populations. Both the incidence and the disease severity were more common among the residents of non-urban environments. Due to the predominantly self-limiting nature of primary dengue infection and folk medicine practices of non-urban populations, we observed a greater number of secondary dengue cases than primary dengue cases. Hemorrhagic manifestations were more in secondary dengue in particularly in the pediatric group. Through different computational methods, ligands RGBLD1, RGBLD2, RGBLD3, and RGBLD4 are proposed as potential inhibitors in silico against DENV-1, -2, -3, and -4 serotypes.

## 1. Introduction

Dengue Virus (DENV) is a member of *Flaviviridae,* mainly spread by Aedes mosquitoes [1]. Currently, there are four DENV serotypes (DENV1-4), sharing 65% genetic homology. DENV infection manifests in various clinical presentations including acute febrile illness, dengue hemorrhagic fever (DHF), dengue shock syndrome (DSS), etc. [2]. Despite DENV posing a global threat, leading to several deaths every year, there is no specific antiviral drug to treat the infection. Several studies have reported inhibitors against dengue envelope protein; however, nearly all studies built the molecules based on the monomer of the envelope protein.

In reality, the dengue envelope is in dimer form, which is an active form of DENV. Only in this dimer state is the dengue virus is capable of infecting its host cells. Moreover, the inhibitor(s) should be stable and exist as an active form, even at high temperatures, to work on the virus at high temperatures of febrile illness. The information about the inhibitor activity at high temperatures is not available, and inhibitors have been studied at a standard temperature in silico. The current study reports the epidemiology of circulating DENV infections. An active form of the envelope protein of DENV-1-4 was built in silico, potential inhibitors were identified, and their stability at febrile condition was studied.

## 2. Materials and Methods

Peripheral venous blood (3 mL) sample was collected by a clinician or a trained phlebotomist within seven days after onset of fever from all eligible patients (as per dengue case definition) [3] included in the study. Patients were attended the hospital from Chittoor, Kadapa, Nellore, Anantapur, East-Godavari, and Guntur districts of Andhra Pradesh state, South India. The serum was aliquoted and stored at −70 °C for later use [4]. Serum was tested for the presence of dengue NS1 antigen using Panbio dengue Early enzyme-linked immunosorbent assay (ELISA) kit (Standard Diagnostics, Inc., Suwon city, Korea). The samples that tested positive for NS1 antigen were tested for dengue IgM (IgM capture ELISA kit, NIV, Pune, India) and IgG (Microlisa, J.Mitra & Co. Pvt. Ltd. New Delhi, India) antibodies following the manufacturer’s instructions.

### 2.1. Dengue Serotyping by rRT-PCR

QIAamp Viral RNA mini kit (Qiagen, Hilden, Germany) was used to purify viral RNA from 140 µL of NS1 antigen-positive serum samples, and RNA was eluted in 60 μL of Buffer AVE. The extracted RNA was reverse transcribed and real-time polymerase chain reaction (rRT-PCR) was performed using Superscript-III one-step RT-PCR kit (Thermo Fisher Scientific Inc., Waltham, MA, USA) using CDC DENV-1-4 Real-Time RT-PCR Kit on Agilent AriaMx Real-Time PCR System (Santa Clara, CA, USA) [5]. The thermal cycling conditions were Stage 1; 30 min at 50 °C, Stage 2; 2 min at 95 °C, and Stage 3; 15 s at 95 °C followed by1 min at 60.0 °C for 45 cycles.

### 2.2. Dengue Envelope Gene Characterization

Two sets of overlapping primers were designed for dengue envelope protein genes of four subtypes (DENV-1, 2, 3, 4) using NCBI primer-BLAST function and reference sequence for each subtype (NC_001477, NC_001474, NC_001475, NC_002640) (Table 1). All rRT-PCR-positive samples were included for the cDNA synthesis using the cDNA Reverse Transcription Kit (Thermo Fisher Scientific, Waltham, MA, USA). Dengue envelop gene PCR was performed with 2 µL of the template (cDNA), each primer (0.8 µM) and 2× PCR mix (Thermo Fisher Scientific). PCR reaction was carried out at 95 °C for 3 min initial denaturation step, 40 cycle of 95 °C/30 s, 55.9 °C or 60.1 °C/30 s, and 72 °C/60 s, and final extension at 72 °C/10 min. The reaction conditions for all four (DENV-1, 2, 3, 4) genotypes were the same except for the annealing temperature (Table 1). PCR products were analyzed by 2% agarose gel electrophoresis, purified and sequenced (Sanger-sequencer, Macrogen Inc., Seoul, Korea).

The acquired sequences were checked for sequence similarity using nucleotide BLAST. The serotypes were interpreted based on the BLAST results. The sequenced amplicons were trimmed using MEGA7 software [6] and submitted to NCBI GenBank. Accession numbers were obtained after submission. The sequences were checked for dengue genotypes and phylogenetic analysis was performed using DENGUE VIRUS TYPING TOOL [7].

### 2.3. Homology Modelling

The nucleotide sequences were translated into amino acids and protein sequences were made using Molecular Evolutionary Genetics Analysis 7 (MEGA7) [6]. A consensus protein sequence was drawn using MEGA7 for each serotype (DENV-1, -2, -3, -4) from the processed sequences after genotyping. The envelope protein was constructed using a template that was searched through National Center for Biotechnology Information (NCBI) (NCBI Resource Coordinators) Basic Local Alignment Search Tool protein (BLASTp) [8] against protein data bank (PDB) [9] based on query coverage and identity. Target template pairwise sequence alignment was performed using CLUSTALX v2.1 [10] and 3D structures for envelope protein of dengue serotypes (homodimer consensus) were built using Modeller v9.21 [11]. Among the generated models for each dengue envelope (DENV) serotype, the best model was selected based on DOPE score. The best models of DENV serotypes were validated through PROCHECK [12], protein structural analysis (ProSA) [13] and protein quality (ProQ) [14] analysis to define the stereo-chemical quality and overall quality of the protein model, respectively.

### 2.4. Protein Processing

The best three-dimensional model structures of DENV-1, -2, -3, -4 active forms (homodimers) were imported to Maestro v11.1 [15] preprocessed by Prime [16] optimized by Epik [17] and energy minimized by using optimized potentials for liquid simulations (OPLS_3) [18] force field with default parameters of protein preparation wizard options of Schrodinger Software Suite. A grid was generated on the interfacial residues on each DENV serotype.

#### 2.4.1. Ligand Processing and Docking of Published Inhibitors

Published inhibitors of DENV were retrieved from literature and were drawn using Marvinsketch 19.3 and/or retrieved from PubChem [19,20,21,22,23]. LigPrep application with inbuilt Epik and OPLS_3 forcefield module of Schrodinger was applied for ligand preparation. Grid-based ligand docking with energetics (GLIDE) [24] extra precision (XP) docking procedure has opted for docking prepared ligands over grids to analyze the binding affinity between the protein and ligand.

#### 2.4.2. Virtual Screening and Docking of Analogs

Each best docked published inhibitor of DENV serotypes based on XPG score was screened for analogues against an in-house library containing more than 28 million compounds. Analogues obtained were docked on the same grids and a similar docking protocol was implemented and the best-scored molecules better than published inhibitors were proposed as leads [25].

#### 2.4.3. Molecular Dynamics Simulations

The interactions and stability of the leads over published inhibitors with each DENV protein subtypes were further assessed by running 100 ns molecular dynamics simulations in Dipalmitoyl phosphatidylcholine (DPPC) membrane for transmembrane residues and simple point charge (SPC) water solvent model system at the febrile condition of 313.15 K were carried by using Desmond’s default protocol [26]. Atomic coordinate data were recorded for every 100 ps as a trajectory and system energies were logged for every 2 fs as a trajectory thereby several molecular dynamics parameters were assessed between published inhibitors and leads with DENV serotypes.

## 3. Results

A total of 3926 patients clinically suspected to have dengue fever (as per the dengue case definition) were interviewed to obtain relevant clinical data and blood samples.

### 3.1. Dengue NS1, IgM, and IgG Enzyme Linked Immunosorbent Assay (ELISA)

Of these 3926 cases, 1316 cases (fever duration < 5 days) were subjected to Dengue nonstructural protein 1 (NS1) antigen enzyme-linked immunosorbent assay (ELISA) following the kit manufacturer’s instructions. Among these, close to 27% (*n* = 354) samples positive for dengue NS1 antigen. Of 354 NS1 positives, 227 (64.1%) and 192 (54.2%) samples tested positive for dengue Immunoglobulin M (IgM) and Immunoglobulin G (IgG) (secondary dengue) antibodies, respectively. Secondary dengue cases (54.2%) were more common than the primary dengue cases (45.8%). Pediatric (≤18; 194 (54.8%)) and rural (rural; 228 (64.4%)) populations were predominantly reported among positive cases. However, there was no significant gender preponderance among dengue NS1-positive cases.

### 3.2. Dengue Serotyping by Real-Time Reverse Transcriptase Polymerase Chain Reaction (rRT-PCR)

All 354 dengue NS1-positive samples were subjected to serotyping by Centre for Cisease Control (CDC, Atlanta, GA, USA) rRT-PCR kit. Multiple serotypes were detected in 18.4% of cases and a total of 449 dengue serotypes were obtained from 354 dengue NS1-positive samples. DENV-2 was the major serotype (191 (42.5%)) followed by DENV-1 (102 (22.7%)), DENV-4 (96 (21.4%)), and DENV-3 (60 (13.4%)). Nearly one-fifth (18.4%) of cases had coinfections with more than one serotype. DENV-1 and -2 serotypes were predominantly observed in co-infections. Notably, 10 pediatric cases had co-infections with all four serotypes and all were primary dengue cases; no such findings were observed in adult cases. Multiple infections were more common among non-urban residents. In secondary dengue, no co-infections other than the co-infection with 1 or 2 were detected (Table 2). All four serotypes were co-existing in the community throughout the year. However, seasonal peaks were observed during rainy months (July and October) (Figure 1).

### 3.3. Clinical Manifestations

As febrile illness was the main inclusion criteria to recruit cases, it was found in all cases. Chills and myalgia were most common manifestations after a febrile illness. Hemorrhagic manifestations were observed in 142 (41.5%) of 354 NS1-positive patients. There was no significant difference in hemorrhagic manifestations between adults (20.6%) and pediatric populations (20.9%). In primary dengue hemorrhagic manifestation was high in the adult population with 23.5% cases, whereas in the pediatric population, it was 14.8%. In secondary dengue hemorrhagic manifestations were high in the pediatric population with 26.0% cases, whereas in adults, it was 18.2% (Table 3a,b).

#### 3.3.1. Dengue Envelope Gene Characterization

Two sets of primers were used to amplify the dengue envelop gene (~1657 bp; coordinates; 853–2509 bp). Target was applied from all samples successfully and around ~920 bp (907–939 bp) bands were observed after 2% agarose gel electrophoresis. The purified product was used for bidirectional sequencing by a sanger sequencer. All samples were successfully sequenced, and the resulting DNA sequence (forward and reverse) and chromatogram were analyzed for any errors/mismatches. The sequences were submitted in the GenBank database for public access (Appendix A).

The FASTA sequences of the DENV-1 to 4 envelope protein genes were used as queries in the dengue virus typing tool [27], and the specific genotypes of dengue were elucidated. Among Dengue subtype-1 (*n* = 109), genotype I (29.4%) and related genotype (23.5) were identified as predominant types followed by genotype-IV-related genotype but not part of genotype IV (25.5%). For three sequences, genotype could not be assigned (Appendix A). DENV-2 Genotype II—Cosmopolitan related (but not part of DENV-2) type was commonly (75.9%) observed in DENV-2 subtype (*n* = 191), followed by genotype V—Asian I (13.9%). As compared to DENV-1, untypable genotypes were greater (8.9%) in the DENV-2 subtype (Appendix A). All DENV-3 samples (*n* = 60) were identified as genotype-III (Appendix A). Genotype II (62.5%) was observed as a dominant genotype under the DENV-4 (*n* = 96) (Appendix A) (Table 4).

#### 3.3.2. Homology Modelling

Hundred models for each of the four consensus DENV serotype dimeric forms were built based on the target–template alignment and atomic coordinates. The best models of DENV serotypes with the least Discrete Optimized Protein Energy (DOPE) score were selected and validated. PROCHECK analysis revealed that ~96.5% of the best model DENV serotypes were in allowed regions of Ramachandran plot, ProSA ‘Z’ score analysis stated that the best models generated were of good quality with ~−5.6, and the predicted LG score of ProQ analysis inferred that the best models were of extremely good models with ~−11.153 (Table 5 and Figure 2, Figure 3, Figure 4 and Figure 5).

#### 3.3.3. Protein Preparation

The three-dimensional structures of the DENV-1, -2, -3, -4 best models were pre-processed by adding the parameters such as added hydrogen atoms, bond order and formal charge corrections, removed atomic clashes, tautomeric alterations, and ionization states of the protein. Finally, the proteins were optimized and had minimal energies. Grid center for DENV-1 (X: 98.803, Y: 6.541, and Z: 197.114), DENV-2 (X: −34.013, Y: −33.357, and Z: 217.829), DENV-3 (X: −132.510, Y: −121.445, and Z: 120.460), and DENV-4 (X: 37.230, Y: −91.139, and Z: 198.825) with an extension of 10 Å from each co-ordinate center (X, Y, and Z) of minimized DENV serotypes for docking of ligands.

#### 3.3.4. Ligand Processing and Docking of Published Inhibitors

LigPrep-processed 93 inhibitors of DENV serotypes with optimal minimal energies for docking on the generated DENV grids. Among the 93 inhibitors the best docked against DENV serotypes are Agnuside with DENV-1 (XPG score of −7.881 kcal/mol), Rhodiolin with DENV-2 (XPG score of −5.1 kcal/mol), Chlorogenic acid with DENV-3 (XPG score of −8.723 kcal/mol), and NITD448.1 with DENV-4 (XPG score of −6.889 kcal/mol) (Figure 2, Figure 3, Figure 4 and Figure 5).

#### 3.3.5. Virtual Screening and Docking of Analogs

Screening of each best-docked compound and Ribavirin (best docked against wild DENV forms) over 28 million compounds resulted in a total of 1117 analogues (Agnuside: 249, Rhodiolin: 333, Chlogenic acid: 249, NITD448.1: 11, and Ribavirin: 275). Docking of 1117 analogues on each grid of DENV serotypes generated 4 leads with better scoring functions and ADME than the existing 95 antivirals (Agnuside analogue for DENV-1 XPG score: −8.675 kcal/mol, Ribavirin analogue for DENV-2 XPG score −9.632 kcal/mol, Chlorogenic acid analogue for DENV-3 XPG score: −8.873 kcal/mol, and Ribavirin analogue for DENV-4 XPG score: −8.772 kcal/mol). The interactions of best docked published inhibitors and leads were depicted in the Figure 2, Figure 3, Figure 4 and Figure 5.

#### 3.3.6. Molecular Dynamics Simulations

The MD simulations studies revealed the conformational stability of RGBLD1-DENV-1; RGBLD2-DENV-2; RGBLD3-DENV-3; RGBLD4-DENV-4 was much more consistent than Agnuside-DENV-1; Rhodioloin-DENV-2; Chlorogenic acid-DENV3; and NITD448-DENV-4 envelope proteins. Molecular dynamic parameters such as root mean square deviations, root mean square fluctuations, energies (total and potential energies of the system), and protein–ligand contacts are much more favorable to the proposed leads than the existing ligands with DENV serotypes (Figure 2, Figure 3, Figure 4 and Figure 5 and Table 6, Table 7, Table 8 and Table 9).

## 4. Discussion

In India, the first dengue fever (DF) case was reported from Madras, Tamil Nadu, in 1946 [28]; later in 1963, an outbreak of DHF was reported from Calcutta, West Bengal [29]. About 33% of global dengue infections were represented by India [30]. Despite frequent outbreaks, there is a paucity in the information concerning circulating genotypes/serotypes from Andhra Pradesh. This is the first study from the state of Andhra Pradesh after the 1965 Visakhapatnam dengue outbreak [31,32].

A total of 1316 samples were included in the study. Of these, 354 samples tested reactive to dengue NS1 antigen. The proportion of male pediatric cases from rural areas was observed to be high with a median age of 15 years (6.7–29.2 years). Of these, 354 dengue NS1-positive cases, 162 (45.8%) were primary infections, and 192 (54.24%) were secondary infections. Among primary infections, the pediatric group was affected most commonly, with a median age of 16 years (7–31 years). The primary cases were reported more in females and rural areas. In secondary infections, pediatric age group and rural areas were also most commonly affected with a median age of 15 years (6–27 years). Males were infected more in secondary infections. In primary cases, DENV-2 was more common, followed by DENV-3, DENV-1/2, DENV-1, and DENV-1/2/3/4. Among the secondary cases, DENV-2 was observed to be more common followed by DENV-4, DENV-1, DENV-3, and DENV-1/2 (Table 2).

In the current study, the pediatric age group was most commonly affected which is in discordance with the other reports from India [33,34,35]. Though the difference was not significant, minimal male preponderance was noted in our study, which is in line with other studies from India [33,36,37,38]. The most common symptom in all cases was fever; myalgia, arthralgia, diarrhea, and vomiting were the other clinical symptoms. Hemorrhagic fever was observed in all serotypes except in coinfections of DENV-1/2/3, DENV-1/3/4, DENV-2/3 with the highest in DENV-2 followed by DENV-4. Among all infections, the DENV-2 infections were more followed by DENV-1, DENV-4, and DENV-3 in decreasing order. In single infections, DENV-2 was the commonest followed by DENV-4, DENV-1, and DENV-3 in declining order. The pediatric age group, males, and rural populations were predominantly contributed to the dengue positive rate in the present study.

Among coinfections, DENV-1/2 coinfections were commonest followed by 1/2/3/4, 2/3, 1/4, 1/2/3 and 1/3/4, 2/4, 1/2/4, 1/3, and 3/4. The pediatric group had a greater number of coinfections. Cases from rural areas and the female gender exhibited a higher number of coinfections (Table 2). India being a hyperendemic country, outbreaks are common. DENV-2 was the sole serotype all through the years 1970–2000 and also resulted in a major outbreak in the year 1996 [39]. DENV-3 slowly replaced DENV-2; thereafter, from 1970 onwards, other serotypes also started circulating with DENV-4 as a rare serotype [32].

The prevalence of dengue subtypes changes from time to time; during 2003–2009, DENV-3 was the major cause of dengue followed by DENV-2 in 2010–2011 and DENV-1 in 2012, which were reported from the northern states of India [35,37]. A similar trend was observed from other parts of India with no or very low prevalence of DENV-4 [29,30,33,38,40,41,42,43]. In contrary to the above reports, DENV-4 was reported as a predominant serotype from the state of Telangana during the 2007 dengue outbreak [44]. Very few studies were conducted on concurrent infections [34,38,45], and the data are scarce concerning serotypes in Andhra Pradesh despite regular outbreaks [30]. In our study, all four serotypes were reported with DENV-2 (41%) preponderance and all possibilities of coinfections were found (Table 2).

A large multicentric study (2018–2019) [46] from India reported a high proportion of secondary dengue cases (65.0%) from southern states, while very low prevalence (<10%) from the northern states of India. The present study also reports relatively high proportion of secondary dengue cases (54.2%) but less prevalence as compared to the above study. This could be due to the inclusion of only NS1 positive samples for analysis in the present study. Five decades back, different dengue serotypes were limited to particular geographies. In recent times, due to the increase in transportation, the migration and evaluation of dengue virus to adapt to new mosquito species outbreaks and co-infections with multiple dengue serotypes have become endemic and have been reported from almost all parts of India [47,48,49]. Compared to North India (more winter months), South India climate is favorable to mosquito breeding and development.

The Indian climate conditions varies widely from very cold Northern hilly states (−14 °C) to very hot Western Indian states (maximum (+)45 °C), predominantly dependent on the Indian monsoon system. The extrinsic incubation period (EIP) of dengue virus largely depends the climatic conditions and reported as short as 5.6 days at +35 °C and as long as 96.5 days at 0 °C. Moderate-to-hot temperate zones (south, central, and western Indian states) favor the mosquito breeding rate and shortening the virus incubation time, thereby increasing the dengue risk and rate of transmission [48]. In the present study and studies from other part of India reported seasonal peaks during the July to October months. During this season, the average climatic temperatures were maintained around 30 to 35 °C with intermittent rains. Furthermore, open drainage system, stagnant of water during rainy seasons, and storage of water in wide mouth earthen pots or reservoirs without lid (act as a breeding ground) could have been contributed to the high prevalence of dengue in this part of the country. In line with the findings of this study, a recent study from India has compiled all dengue outbreaks in India since the last 50 years, and the authors reported that most dengue outbreaks occurred predominantly during the monsoon (June to September) or post-monsoon (October to December) period [41]. These findings support the relation between dengue seasonality and roles of both rainfall and ambient temperature.

In line with the above findings, recent reports from southern states observed high disease burden and has led to an increase in secondary dengue cases. Due to the open drainage system, poor mosquito prevention, and hygiene practices and awareness, the present study observed high prevalence of dengue cases in rural areas. This is an important information required for implementing effective prevention and clinical management protocols in the hotspot regions [50].

Out of 93 small molecules published against the dengue envelope protein, Agnuside, Rhodiolin, Chlorogenic, and NITD448 showed good interactions with the envelope protein models for DENV-1, -2, -3, and -4, respectively. The analogues of these leads which showed good interactions with envelope protein were RGBLD 1, 2, 3, and 4. Grid region residues showed good interactions with both chains of dimers in DENV-1, -2, -3, and -4.

In 100 ns MDS, Agnuside interacted with 29 amino acids, whereas RGBLD1 interacted with 33 amino acids with high interaction fraction compared to Agnuside in DENV-1. Rhodiolin showed interactions with 21 amino acids, while RGBLD2 displayed interactions with 34 ammino acids in DENV-2. Chlorogenic acid showed interactions with 27 amino acids, whereas RGBLD3 interacted with 43 amino acids of DENV-3. NITD448 exhibited interactions with 30 amino acids, while RGBLD4 interacted with 60 amino acids of DENV-4.

The important amino acids which showed stable interactions in the RGBLD1/DENV-1envelope protein were Phe90 and Asp235 in the A chain and Ala88, Phe90, Gln234, Asp235 in the B chain. Stable interactions were seen between RGBLD2/DENV-2, Asn10, Ile24, Glu31, Leu32, Val432, and Gly433 in A chain. In RGBLD3/DENV-3, there were stable interactions with Arg231 in A chain and with Glu83, and His92 in the B chain. In RGBLD4/DENV-4, His230, Glu84, and Tyr90, there were stable interactions throughout the simulations. These analogues showed interactions with both the chains of the dengue envelope protein dimer. The pharmacological properties of the best leads were correlated favorably with more than 95% of approved drug molecules.

The study focused on 93 published inhibitors to dock with the respective DENV serotype envelope protein obtained from the study samples; as a result, the four best molecules were shortlisted for further use. Among 93 published inhibitors, Agnuside was found to be the best inhibitor for DENV-1, Rhodiolin for DENV-2, Chlorogenic acid for DENV-3, and NITD448 for DENV-4. Agnuside and Rhodiolin were reported elsewhere as the best molecules for dengue virus protease (NS2B-NS3pro), helicase (NS3 helicase), methyltransferase (MTase), and RdRp of DENV serotypes [51].

Similarly, Ribavirin was found to be effective against RdRp of DENV and other flaviviruses [52]. Chlorogenic acid is another broad-spectrum antiviral molecule found to be active against influenza A (H1N9) virus RdRp and neuraminidase (NA), and chikungunya virus glycoprotein (E3-E2-E1) and protease (nsP2) [53]. In our study, during in silico analysis, these molecules showed the best binding scores when docked with our protein construct. Using these structures as a reference molecule, we searched for the novel and potent analogs against 285 million compounds. Based on the scores obtained, RGBLD1 (Agnuside analog), RGBLD2 and RGBLD4 (Ribavirin analogs), and RGBLD3 (Chlorogenic acid analog) were found as the best analogs, and all new molecules scored better than its analogs. Hence, the new analogs (RGBLD-1 to 4) would be a potential and broad-spectrum antiviral candidate against dengue serotypes and other similar viruses, as shown for its reference molecules. However, further in vitro and in vivo model studies are required to validate these new molecules.

Altogether, the best leads showed satisfactory interactions with a greater number of amino acids with two chains of the envelope protein, interactions fraction, and pharmacological properties. Strong interactions between the dengue envelope proteins and leads constrict the movements of homodimers of the envelope, thus stopping the viral entry into the host. Therefore, it can be proposed that the best leads can be potential inhibitors of the dengue virus.

## Figures and Tables

**Figure 1 viruses-14-00940-f001:**
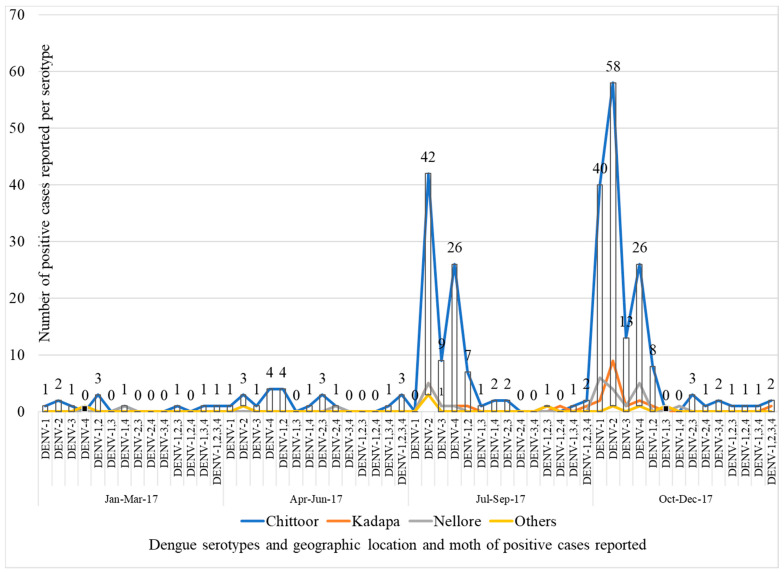
Trends of occurrence of different Dengue virus serotypes and their co-infections in the given geographic location during January 2017 to December 2017. Dengue cases were occurring throughout the year, with seasonal peaks in July and October months. During outbreaks all four dengue serotypes were observed with DENV-2 predominance. There were no hotspot regions observed during the study period in this (Raayalaseema region, Andhra Pradesh, India) part of the state.

**Figure 2 viruses-14-00940-f002:**
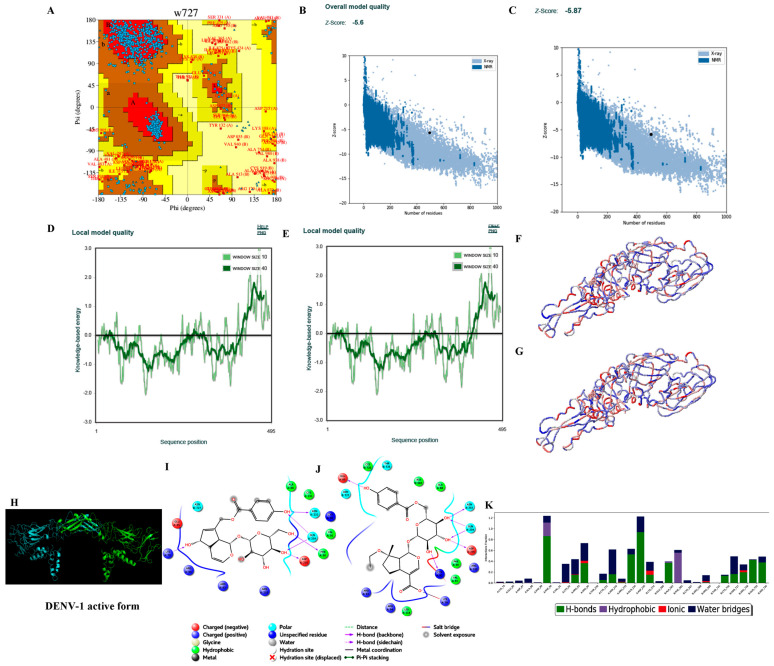
Structure validation of DENV-1 by various in silico methods. (**A**) PROCHECK analysis (Ramachandran plot of the best DENV-1 model). (**B**) Model quality before minimization (ProSA). (**C**) Model quality after minimization (ProSA). (**D**) Per residue model quality before minimization. (**E**) Per residue model quality after minimization. (**F**) Ribbon form of DENV-1 structure before minimization (blue color—least energy regions; red color—highest energy regions). (**G**) Ribbon form of DENV-1 structure after minimization (blue color—least energy regions; red color—highest energy regions). Structure of DENV-1 envelop protein and docking and dynamics interactions of reference and novel inhibitor molecules. (**H**) Cartoon representation of the best DENV-1 dimer model (A chain in green color and B chain in cyan color). (**I**) Best docked published inhibitor (Agnuside) interactions with DENV-1. (**J**) Best docked lead 1 (RGBLD-1) interactions with DENV-1; RGBLD1 showed interactions with 17 amino acids. Six bonded interactions (Hydrogen-5, Salt bridge-1) were found. Glu85, Asn232, Gln234, Asp235, Cys92 showed hydrogen bonding with RGBLD1; Lys79 showed salt bridge interaction. Non-bonding interactions include polar, hydrophobic, positive charge, and negative charge with amino acids Ala88, Phe90, Val91, Leu114, Arg94, Arg93 of chain A and Asn727, Cys726, Thr725, Ala583 of chain B of DENV-1. (**K**) DENV-1-RGBLD-1 (protein ligand contacts) in 100 ns molecular dynamics simulations.

**Figure 3 viruses-14-00940-f003:**
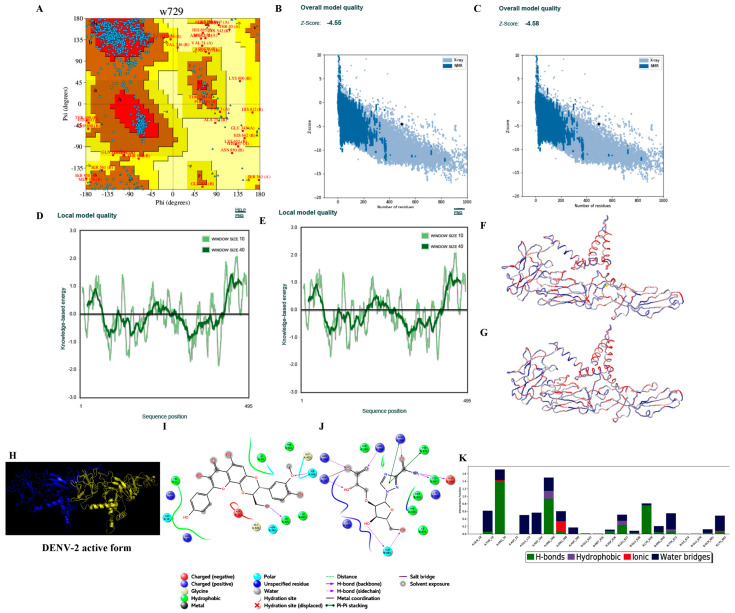
Structure validation of DENV-2 by various in silico methods. (**A**) PROCHECK analysis (Ramachandran plot of the best DENV-2 model). (**B**) Model quality before minimization (ProSA). (**C**) Model quality after minimization (ProSA). (**D**) Per residue model quality before minimization. (**E**) Per residue model quality after minimization. (**F**) Ribbon form of DENV-2 structure before minimization (blue color—least energy regions; red color—highest energy regions). (**G**) Ribbon form of DENV-2 structure after minimization (blue color—least energy regions; red color—highest energy regions). Structure of DENV-2 envelop protein and docking and dynamics interactions of reference and novel inhibitor molecules. (**H**) Cartoon representation of the best DENV-2 dimer model (A chain in blue color and B chain in yellow color). (**I**) Best docked published inhibitor (Rhodiolin) interactions with DENV-2. (**J**) Best docked lead 1 (RGBLD-2) interactions with DENV-2; In DENV-2, RGBLD2 showed interactions with 14 amino acids. Ten bonded interactions were found consisting of hydrogen bonding with Arg288, Arg286, Arg20 in chain A; Leu342, Glu343, Gln386, Arg345 in chain B. Pi–pi stacking was observed in Arg288 in chain A, Tyr377 in chain B. Salt bridge with Arg20 was observed in chain A. Non-bonding interactions include polar, hydrophobic, positive charge, and negative charge with amino acids Phe186, Phe169 of chain A; Lys344, Met340, Ile379, Lys388 of chain B in DENV-2. (**K**) DENV-2-RGBLD-2 (protein ligand contacts) in 100 ns molecular dynamics simulations.

**Figure 4 viruses-14-00940-f004:**
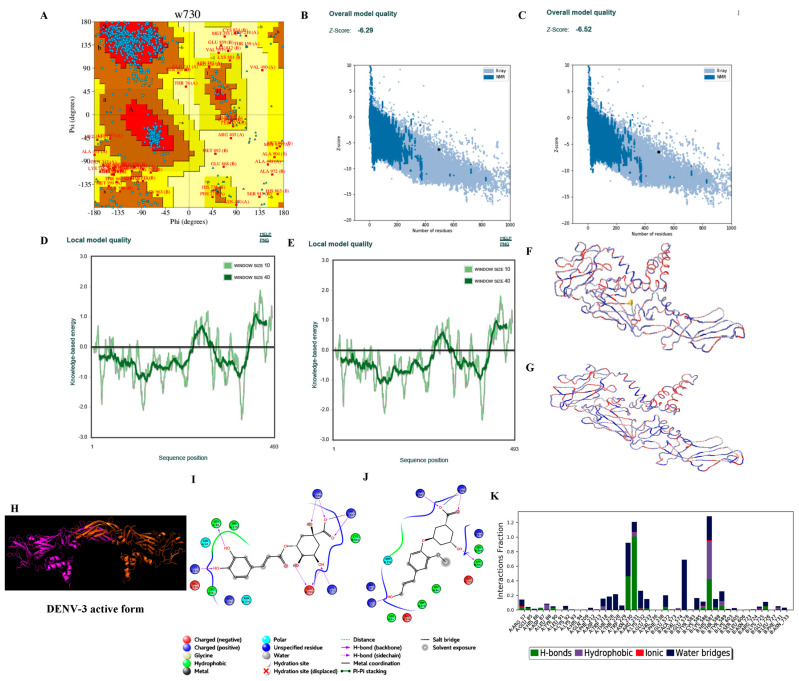
Structure validation of DENV-3 by various in silico methods. (**A**) PROCHECK analysis (Ramachandran plot of the best DENV-3 model). (**B**) Model quality before minimization (ProSA). (**C**) Model quality after minimization (ProSA). (**D**) Per residue model quality before minimization. (**E**) Per residue model quality after minimization (**F**) Ribbon form of DENV-3 structure before minimization (blue color—least energy regions; red color—highest energy regions). (**G**) Ribbon form of DENV-3 structure after minimization (blue color—least energy regions; red color—highest energy regions). Structure of DENV-3 envelope protein and docking and dynamics interactions of reference and novel inhibitor molecules. (**H**) Cartoon representation of the best DENV-3 dimer model (A chain in magenta color and B chain in red color). (**I**) Best docked published inhibitor (Chlorogenic acid) interactions with DENV-3. (**J**) Best docked lead 1 (RGBLD-3) interactions with DENV-3; In DENV-3, RGBLD3 showed interactions with 13 amino acids. Five bonded interactions were found comprising of hydrogen bonding with Lys232 of chain A; His92, Lys91, Cys90 of chain B. Salt bridge was observed in Lys91 of chain B. Non-bonding interactions include polar, hydrophobic, positive charge, and negative charge with amino acids Leu88, Trp229, Glu233, Leu91, Arg231, Asn230 of chain A; Leu89, Glu231 of chain B. (**K**) DENV-3-RGBLD-3 (protein ligand contacts) in 100 ns molecular dynamics simulations.

**Figure 5 viruses-14-00940-f005:**
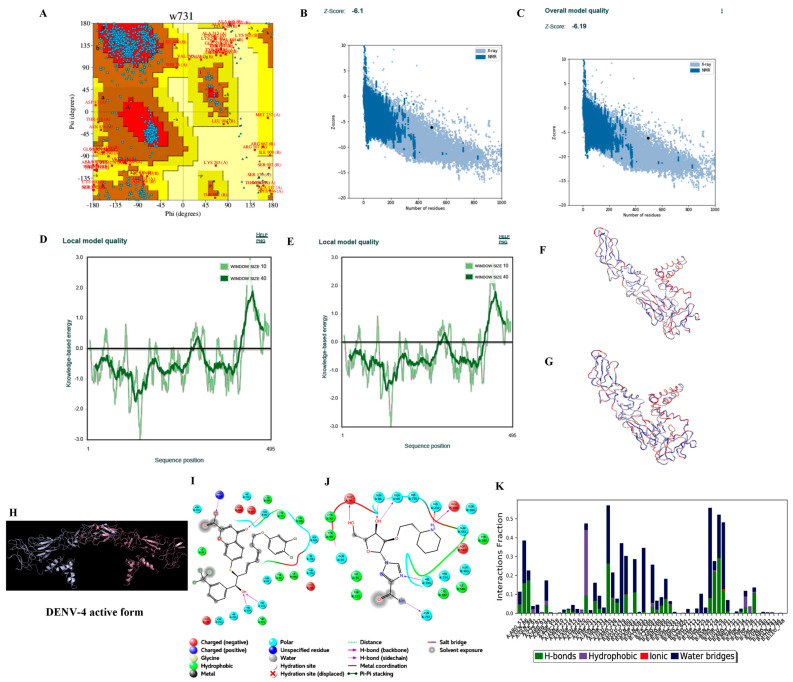
Structure validation of DENV-4 by various in silico methods. (**A**) PROCHECK analysis (Ramachandran plot of the best DENV-4 model) (**B**) Model quality before minimization (ProSA) (**C**) Model quality after minimization (ProSA) (**D**) Per residue model quality before minimization. (**E**) Per residue model quality after minimization (**F**) Ribbon form of DENV-4 structure before minimization (blue color—least energy regions; red color—highest energy regions). (**G**) Ribbon form of DENV-4 structure after minimization (blue color—least energy regions; red color—highest energy regions). Structure of DENV-4 envelop protein and docking and dynamics interactions of reference and novel inhibitor molecules. (**H**) Cartoon representation of the best DENV-4 dimer model (A chain in silver color and B chain in brown color). (**I**) Best docked published inhibitor (NITD448) interactions with DENV-4. (**J**) Best docked lead 1 (RGBLD-4) interactions with DENV-4; In DENV-4, RGBLD4 showed interactions with 21 amino acids. Seven hydrogen bonding were found with Glu85, Gln86, Gln88, His230, His233, Asn232 of chain A; Glu85 of chain B. Non-bonding interactions include polar, hydrophobic and negative charge with amino acids Thr234, Trp231, Ile91, Gln89, Tyr90, Cys92 of chain A; His230, Gln89, Gln86, Tyr90, Asp87, Gln88, Ile91, Cys92 of chain B. (**K**) DENV-4-RGBLD-4 (protein ligand contacts) in 100 ns molecular dynamics simulations.

**Table 1 viruses-14-00940-t001:** Primer sequences and Annealing Temperatures of DENV 1-4 envelope PCR.

Subtype	Primer	Sequence (5′-3′)	Position	Target Size (bp)	Annealing Temp.
First Half (FH) primers					
DENV-1	Forward	TCTAGCACATGCCATAGGAACA	853–874	917	55.9 °C
	Reverse	AAATTGTTGTCGTTCCAGACGTT	1769–1749		
DENV-2	Forward	GGCATACACCATAGGAACGACA	858–879	923	55.9 °C
	Reverse	GTCCTGTGAAGAGTAAGTTTCCTGA	1780–1756		
DENV-3	Forward	ACTAGCCCTATTTCTCGCCCA	8411–861	939	60.1 °C
	Reverse	TTTAAGTGCCCCGCGAAAATG	1779–1759		
DENV-4	Forward	CGCTCTTGGCAGGATTTATGG	841–861	927	55.9 °C
	Reverse	GATTTCCATCACCGGAGTCCA	1767–1747		
Second Half (SH) primers					
DENV-1	Forward	GGGGGCTTCAACATCCCAAG	1600–1619	910	55.9 °C
	Reverse	CTCTGTCCAGGTGTGGACTTC	2509–2489		
DENV-2	Forward	CCGGAGCGGACACACAAG	1601–1618	907	60.1 °C
	Reverse	GTCCATGTGTGCACGTTGTCT	2507–2487		
DENV-3	Forward	ACAGAAACACCAACCTGGAACA	1604–1625	918	55.9 °C
	Reverse	TGCTTGGAATTTGTATTGCTCTGT	2521–2498		
DENV-4	Forward	AGCAGGAGCAGACACATCAGA	1601–1621	921	55.9 °C
	Reverse	TTGTACTGTTCTGTCCAAGTGTGC	2521–2498		

**Table 2 viruses-14-00940-t002:** Distribution of dengue serotypes in primary and secondary dengue cases and socio-demographic characteristics.

Serotype (*n*)	Case Type (*n*)	Age ≤ 18 Years	Age > 18 Years	Male	Female	Urban	Rural	*p*-Value
1 (50)	primary (10)	7	3	4	6	4	6	0.0002 *
	secondary (40)	19	21	22	18	14	26	
2 (140)	primary (74)	28	46	31	43	34	40	0.574
	secondary (66)	36	30	38	28	23	43	
3 (30)	primary (14)	10	4	9	5	7	7	0.621
	secondary (16)	8	8	11	5	2	14	
4 (69)	primary (10)	4	6	8	2	3	7	0.0001 *
	secondary (59)	36	23	31	28	23	36	
1 and 2 (24)	primary (13)	3	10	7	6	3	10	0.607
	secondary (11)	10	1	4	7	1	10	
1, 2, and 3 (4)	primary (4)	4	0	0	4	2	2	0.006 *
	secondary (0)	0	0	0	0	0	0	
1, 2, 3, and 4 (10)	primary (10)	10	0	5	5	2	8	0.002 *
	secondary (0)	0	0	0	0	0	0	
1, 2, and 4 (2)	primary (2)	2	0	2	0	1	1	0.006 *
	secondary (0)	0	0	0	0	0	0	
1 and 3 (2)	primary (2)	1	1	1	1	0	2	0.009 *
	secondary (0)	0	0	0	0	0	0	
1, 3, and 4 (4)	primary (4)	3	1	3	1	3	1	0.0004 *
	secondary (0)	0	0	0	0	0	0	
1 and 4(6)	primary (6)	4	2	1	5	0	6	0.003 *
	secondary (0)	0	0	0	0	0	0	
2 and 3(8)	primary (8)	7	1	5	3	3	5	0.0003 *
	secondary (0)	0	0	0	0	0	0	
2 and 4(3)	primary (3)	1	2	1	2	0	3	0.002 *
	secondary (0)	0	0	0	0	0	0	
3 and 4 (2)	primary (2)	1	1	1	1	1	1	3.759
	secondary (0)	0	0	0	0	0	0	

Significance levels are indicated by the superscripts ** p* ≤ 0.05.

**Table 3 viruses-14-00940-t003:** (**a**) Signs and symptoms recorded in primary and secondary dengue infections and pediatric and adult populations. (**b**) Signs and symptoms recorded in primary and secondary dengue infections and pediatric and adult populations.

(a)												
Serotypes	Case Type (*n*)	Feb%	Chi%	Mya%	Head%	Retro%	Arth%	Mal%	Rig%	Hem%	Rash%	*p*-Value
1 (*n* = 50)	1° dengue (10)	100	80	90	80	-	10	20	10	30	-	
	2° dengue (40)	100	85	73	48	13	25	23	23	50	5	0.006 *
	≤18 years (26)	100	81	77	50	4	23	19	15	46	4	
	>18 years (24)	100	88	75	58	17	21	25	25	46	4	0.001 *
	all cases (50)	100	84	76	54	10	22	22	20	46	4	
2 (*n* = 140)	1° dengue (74)	100	80	88	58	12	30	20	16	43	7	
	2° dengue (66)	100	82	77	50	15	18	12	24	38	11	0.6
	≤18 years (64)	100	84	81	42	8	14	14	17	36	14	
	>18 years (76)	100	78	84	64	18	33	18	22	45	4	0.39
	all cases (140)	100	81	83	54	14	24	16	20	41	9	
3 (*n* = 30)	1° dengue (14)	100	93	71	86	7	43	29	21	21	7	
	2° dengue (16)	100	81	88	63	6	38	-	19	19	6	0.999
	≤18 years (18)	100	78	72	67	6	28	17	17	17	6	
	>18 years (12)	100	100	92	83	8	58	8	25	25	8	0.6
	all cases (30)	100	87	80	73	7	40	13	20	20	7	
4 (*n* = 69)	1° dengue (10)	100	80	90	80	20	40	20	20	50	10	
	2° dengue (59)	100	81	75	46	12	25	19	14	49	5	0.005 *
	≤18 years (40)	100	83	70	38	8	10	15	10	48	8	
	>18 years (29)	100	79	86	69	21	52	24	21	52	3	0.8
	all cases (69)	100	81	77	51	13	28	19	14	49	6	
1/2 (*n* = 24)	1° dengue (13)	100	69	100	69	-	15	15	15	38	8	
	2° dengue (11)	100	100	55	18	18	18	18	18	73	-	0.6
	≤18 years (13)	100	92	69	31	15	8	8	15	62	-	
	>18 years (11)	100	73	91	64	-	27	27	18	45	9	0.9
	all cases (24)	100	83	79	46	8	17	17	17	54	4	
1/2/3 (*n* = 4)	1° dengue (4)	100	100	75	25	-	-	-	-	-	25	
	2° dengue (nil)	-	-	-	-	-	-	-	-	-	-	0.020 *
	≤18 years (4)	100	100	75	25	-	-	-	-	-	25	
	>18 years (nil)	-	-	-	-	-	-	-	-	-	-	0.020 *
	all cases (4)	100	100	75	25	-	-	-	-	-	25	
1/2/3/4 (*n* = 10)	1° dengue (10)	100	80	80	30	-	-	-	-	40	-	
	2° dengue (nil)	-	-	-	-	-	-	-	-	-	-	00017 *
	≤18 years (10)	100	80	80	30	-	-	-	-	40	-	
	>18 years (nil)	-	-	-	-	-	-	-	-	-	-	0.017 *
	all cases (10)	100	80	80	30	-	-	-	-	40	-	
(**b**)												
**Serotypes**	**Case Type (*n*)**	**Feb%**	**Chi%**	**Mya%**	**Head%**	**Retro%**	**Arth%**	**Mal%**	**Rig%**	**Hem%**	**Rash%**	***p*-Value**
1/2/4 (*n* = 2)	1° DENGUE (2)	100	100	50	-	-	-	-	-	50	-	
	2° DENGUE (nil)	-	-	-	-	-	-	-	-	-	-	0.037 *
	≤18 years (2)	100	100	50	-	-	-	-	-	50	-	
	>18 years (nil)	-	-	-	-	-	-	-	-	-	-	0.037 *
	All cases (2)	100	100	50	-	-	-	-	-	50	-	
1/3 (*n* = 2)	1° DENGUE (2)	2	100	100	-	-	-	50	-	50	-	
	2° DENGUE (nil)	-	-	-	-	-	-	-	-	-	-	0.013 *
	≤18 years (1)	1	100	100	-	-	-	-	-	-	-	
	>18 years (1)	1	100	100	-	-	-	100	-	100	-	0.388
	All cases (2)	2	100	100	-	-	-	50	-	50	-	
1/3/4 (*n* = 4)	1° DENGUE (4)	4	100	100	25	-	-	-	-	-	-	
	2° DENGUE (nil)	-	-	-	-	-	-	-	-	-	-	0.043 *
	≤18 years (3)	3	100	100	-	-	-	-	-	-	-	
	>18 years (1)	1	100	100	100	-	-	-	-	-	-	0.317
	All cases (4)	4	100	100	25	-	-	-	-	-	-	
¼ (*n* = 6)	1° DENGUE (6)	6	67	83	50	17	33	33	33	83	17	
	2° DENGUE (nil)	-	-	-	-	-	-	-	-	-	-	0.001 *
	≤18 years (4)	4	75	75	25	-	25	25	25	75	25	
	>18 years (2)	2	50	100	100	50	50	50	50	100	-	0.299
	All cases (6)	6	67	83	50	17	33	33	33	83	17	
2/3 (*n* = 8)	1° DENGUE (8)	8	88	50	38	-	13	-	25	-	13	
	2° DENGUE (nil)	-	-	-	-	-	-	-	-	-	-	0.011 *
	≤18 years (7)	7	86	43	29	-	14	-	29	-	14	
	>18 years (1)	1	100	100	100	-	-	-	-	-	-	0.038 *
	All cases (8)	8	88	50	38	-	13	-	25	-	13	
2/4 (*n* = 3)	1° DENGUE (3)	3	67	67	100	-	33	33	-	33	-	
	2° DENGUE (nil)	-	-	-	-	-	-	-	-	-	-	0.002 *
	≤18 years (1)	1	-	-	100	-	-	100	-	-	-	
	>18 years (2)	2	100	100	100	-	50	-	-	50	-	0.051 *
	All cases (3)	3	67	67	100	-	33	33	-	33	-	
3/4 (*n* = 2)	1° DENGUE (2)	2	50	100	-	-	-	-	-	100	-	
	2° DENGUE (nil)	-	-	-	-	-	-	-	-	-	-	0.031 *
	≤18 years (1)	1	-	100	-	-	-	-	-	100	-	
	>18 years (1)	1	100	100	-	-	-	-	-	100	-	0.66
	All cases (2)	2	50	100	-	-	-	-	-	100	-	

Feb: Febrile illness; Chi: Chills; Mya: Myalgia; Head: Headache; Retro: Retro-orbital pain; Arth: Arthralgia; Mal: Malaise; Rig: Rigors; Hem: Hemorrhagic manifestations. Significance levels are indicated by the superscripts * *p* ≤ 0.05.

**Table 4 viruses-14-00940-t004:** Prevalence and distribution of dengue genotypes reported in the present study.

Subtype	Genotype	Total	(%)
1	Could not assign	3	(2.94)
	DENV-1 Genotype I	30	(29.41)
	DENV-1 Genotype IV	4	(3.92)
	Related to but not part of DENV-1 Genotype I	24	(23.53)
	Related to but not part of DENV-1 Genotype I and IV	15	(14.71)
	Related to but not part of DENV-1 Genotype IV	26	(25.49)
	Total	109	
2	Could not assign	17	(8.90)
	DENV-2 Genotype V—Asian I	25	(13.09)
	Related to but not part of DENV-2 Genotype II—Cosmopolitan	145	(75.92)
	Related to but not part of DENV-2 Genotype V—Asian I	1	(0.52)
	Related to but not part of DENV-2 Genotype VI—Sylvatic	3	(1.57)
	Total	191	
3	DENV-3 Genotype III	60	
4	DENV-4 Genotype I	10	(10.42)
	DENV-4 Genotype II	60	(62.50)
	Related to but not part of DENV-4 Genotype II	26	(27.08)
	Total	96	
	Grand total	449	

**Table 5 viruses-14-00940-t005:** Parameters used to build 3D structure of envelop protein from consensus sequence.

S. No.	Parameters	DENV-1	DENV-2	DENV-3	DENV-4
1.	Template (PDB_ID) used to build 3D structure of consensus sequence derived from this study isolates.	4C2I (A and C chains)	1P58 (A and B chains)	3J6S (A and C chains)	4CBF (A and C chains)
2.	Query coverage with template	100%	100%	100%	100%
3.	Consensus sequence identity with template	63.43%	62.22%	88.84%	84.44%
4.	Best model	8th model	27th model	19th model	38th model
5.	DOPE score of the best model	−84,972.195 kcal/mol	−84,226.984 kcal/mol	−87,818.578 kcal/mol	−83,635.703 kcal/mol
6.	Ramachandran plot-residues falling under allowed regions (excluding Gly and Pro residues)	850/872 (97.48%) * 854/872 (97.93%) ^#^	852/866 (98.38%) * 849/866 (94.8%) ^#^	831/848 (97.99%) * 833/848 (98.23%) ^#^	832/856 (97.20%) * 833/856 (97.31%) ^#^
7.	ProSA (Z-score)	−5.60 (−5.87)	−4.58 (−4.55)	−6.29 (−6.52)	−6.10 (−6.19)
8.	ProQ	11.362	11.476	11.454	10.319

* Values before minimization; ^#^ values after minimization.

**Table 6 viruses-14-00940-t006:** Comparison of scores obtained in the molecular dynamic’s simulation parameters for DENV-1 reference molecule and novel analog molecule.

S. No.	Parameters/Properties during 1000 Trajectories of 100 ns MDS	DENV-1_Agnuside	DENV-1_RGBLD1
1.	Total Energy (kcal/mol)	−409,942.737	−410,034.482
2.	Potential Energy (kcal/mol)	−583,489.002	−583,632.685
3.	Degrees of freedom	561,805	561,974
4.	Number of particles	259,876	259,959
5.	Protein-Ligand RMSD: Cα, backbone, sidechain, protein hetero atoms, ligand with regard to protein, ligand with regard ligand (Å)	5.181, 5.180, 6.021, 5.523, 5.556, 0.956	4.954, 4.958, 5.802, 5.314, 6.456, 2.419
6.	Protein RMSF: Cα, backbone, sidechain, protein hetero atoms (Å)	2.447, 2.454, 2.818, 2.627	2.622, 2.632, 3.000, 2.810
7.	Ligand RMSF: ligand with regard to protein, ligand with regard to ligand (Å)	2.582, 0.531	3.514, 1.341
8.	Hydrogen bonds	3608	4900
9.	Hydrophobic interactions	1844	410
10.	Ionic interactions	-	196
11.	Metallic interactions	11	24
12.	Pi–cation interactions	-	4
13.	Pi–pi stacking interactions	57	441
14.	Water bridge interactions	3702	3251
15.	Total number of Interactions	9222	9226

**Table 7 viruses-14-00940-t007:** Comparison of scores obtained in the molecular dynamic’s simulation parameters for DENV-2 reference molecule and novel analog molecule.

S. No.	Parameters/Properties during 1000 Trajectories of 100 ns MDS	DENV-2_Rhodiolin	DENV-2_RGBLD2
1.	Total Energy (kcal/mol)	−384,355.621	−394,785.237
2.	Potential Energy (kcal/mol)	−529,888.872	−542,847.189
3.	Degrees of freedom	470,928	479,089
4.	Number of particles	219,287	223,321
5.	Protein-Ligand RMSD: Cα, backbone, sidechain, protein hetero atoms, ligand with regard to protein, ligand with regard to ligand (Å)	7.855, 7.848, 8.671, 6.743, 2.043	7.591, 7.581, 8.484, 7.973, 4.701, 0.654
6.	Protein RMSF: Cα, backbone, sidechain, protein hetero atoms (Å)	3.190, 3.215, 3.634, 3.423	2.916, 3.047, 3.433, 3.237
7.	Ligand RMSF: ligand with regard to protein, ligand with regard to ligand (Å)	2.653, 0.784	1.699, 0.296
8.	Hydrogen bonds	3209	3762
9.	Hydrophobic interactions	462	165
10.	Ionic interactions	-	342
11.	Metallic interactions	23	2
12.	Pi–cation interactions	95	213
13.	Pi–pi stacking interactions	73	22
14.	Water bridge interactions	4345	4100
15.	Total number of Interactions	8207	8606

**Table 8 viruses-14-00940-t008:** Comparison of scores obtained in the molecular dynamic’s simulation parameters for DENV-3 reference molecule and novel analog molecule.

S. No.	Parameters/Properties during 1000 Trajectories of 100 ns MDS	DENV-3_Chlorogenic Acid	DENV-3_RGBLD3
1.	Total Energy (kcal/mol)	−418,919.728	−419,029.352
2.	Potential Energy (kcal/mol)	−589,515.379	−589,610.425
3.	Degrees of freedom	552,135	552,167
4.	Number of particles	256,125	256,142
5.	Protein-Ligand RMSD: Cα, backbone, sidechain, protein hetero atoms, ligand with regard to protein, ligand with regard to ligand (Å)	5.606, 5.607, 6.465, 5.975, 5.091, 2.382	4.584, 4.589, 5.439, 4.940, 8.760, 1.327
6.	Protein RMSF: Cα, backbone, sidechain, protein hetero atoms (Å)	2.839, 2.848, 3.177, 3.008	2.843, 2.851, 3.194, 3.014
7.	Ligand RMSF: ligand with regard to protein, ligand with regard to ligand (Å)	2.623, 0.978	5.472, 0.943
8.	Hydrogen bonds	2625	2306
9.	Hydrophobic interactions	6	225
10.	Ionic interactions	20	46
11.	Metallic interactions	2	11
12.	Pi–cation interactions	40	353
13.	Pi–pi stacking interactions	1	278
14.	Water bridge interactions	3122	3553
15.	Total number of Interactions	5816	6772

**Table 9 viruses-14-00940-t009:** Comparison of scores obtained in the molecular dynamic’s simulation parameters for DENV-4 reference molecule and novel analog molecule.

S. No.	Parameters/Properties during 1000 Trajectories of 100 ns MDS	DENV-4_NITD448	DENV-4_RGBLD4
1.	Total Energy (kcal/mol)	−401,751.525	−402,105.698
2.	Potential Energy (kcal/mol)	−568,014.934	−568,391.175
3.	Degrees of freedom	538,245	538,175
4.	Number of particles	249,220	249,195
5.	Protein-Ligand RMSD: Cα, backbone, sidechain, protein hetero atoms, ligand with regard to protein, ligand with regard to ligand (Å)	5.775, 5.759, 6.665, 6.123, 8.256, 2.038	4.406, 4.403, 5.417, 4.853, 3.892, 1.834
6.	Protein RMSF: Cα, backbone, sidechain, protein hetero atoms (Å)	2.800, 2.804, 3.194, 2.994	2.569, 2.580, 2.952, 2.761
7.	Ligand RMSF: ligand with regard to protein, ligand with regard to ligand (Å)	8.993, 1.022	2.684, 1.346
8.	Hydrogen bonds	2303	2293
9.	Hydrophobic interactions	111	836
10.	Ionic interactions	11	91
11.	Metallic interactions	26	231
12.	Pi–cation interactions	417	78
13.	Pi–pi stacking interactions	38	415
14.	Water bridge interactions	4269	5286
15.	Total number of Interactions	7175	9230

## Data Availability

Not applicable.

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
