# Peer review of "Molecular Characterization and Identification of Potential Inhibitors for ‘E’ Protein of Dengue Virus"

_viruses, 2022, doi:10.3390/v14050940_

Round 1

Reviewer 1 Report

Dengue virus is one of the most important viruses causing dengue fever, and the development of antiviral agents is one of the most important issues. In this study, in silico method of inhibiting envelope protein dimerization was implemented for drug discovery. In recent years, computational science has been a powerful tool in drug discovery, and many candidates have been developed by such methods. The candidate obtained in this study is no exception and is a promising anti-DENV drug and this study has great importance in this context. The study also reported that many coinfections with several serotypes of DENV were diagnosed in this study. This finding is remarkable, and the report is of great epidemiological importance. This manuscript is well-written and, after addressing the questions and comments below, I believe it is worthy of publication in Viruses.

1, The authors have shown homology modeling in Figures 2-5 and docking simulations in Figures 6-9. We fear that readers may find it difficult to understand if there are separate figures showing similar messages, as is the case in this manuscript. Therefore, I believe Figures 2-5 and 6-9 should be integrated for each serotype. On a different note about Figure, the authors used “in silico” in main text and “in-silico” in figure legend. Please unify.

2, Peripheral venous blood was collected from 3926 individuals in this study. When was the blood drawn? These data would be valuable as epidemiology and I suggest that the information be provided to enhance the value of the data. Also I have a question whether there is a correlation between the results of serotyping and the occurrence in the society. Also, were all the samples taken in India? We inferred this because the author is affiliated with a laboratory in India and discusses the situation of infectious diseases in India. I suggest adding the locations to the Materials and Methods section to clarify that information.

3, The authors performed sequencing and showed DENV genotype and accession number. I recommend that the phylogenetic tree be shown.

4, The authors performed virtual screening and docking simulations for inhibitors against envelope protein. 93 candidates have been reported in other studies so far, but I think in vitro analysis is also necessary to inhibit dimerization. If could, please perform and provide its in vitro result. If this could not be done due to technical problems with the envelope protein dimerization analysis, I suggest discussing the relationship between the results of the inhibition effect and the in silico properties.

5, The authors showed only best docked candidates against each serotype. If could, please show the docking results for the other 92 compounds.

Reviewer 2 Report

The manuscript “Molecular Characterization and Identification of Potential Inhibitors for ‘E’ Protein of Dengue Virus” presents important information regarding the circulating DENV strains at the study location, which adds to the existing knowledge base meaningfully.

It looks like this manuscript presents two weakly linked modules: 1. epidemiological + molecular data, 2. Simulation data for protein/ligand binding. For most readers these two modules are not equally understandable or interesting. For example, I got meaningful information on the DENV epidemiology and serotype/strain information from the robust number of specimens tested, but most of the figures and tables in the molecular docking experiments are incomprehensible, I understand some of the numbers and figures, but don’t understand their significance. So, I strongly suggest to separate them out. The molecular docking simulations are based on existing DENV E protein models from a public database, which is accessible even before getting the sequence data from this manuscript. The current merging of these two modules would have been justified if this were a new virus with very little protein structure information, and the docking simulations would suggest a class of lead molecules. However, the drug design module as a stand-alone manuscript would be interesting to some readers who may find that the stereotypic differences between DENV infections may require difference drugs (or combinations of drugs) to function effectively.

Most of my comments would focus on the first half of the data. The second half would merit a separate manuscript.

--please provide the geographical location of the healthcare facility and patient population. Please describe some general environmental or population factors that may be unique or similar compared to other DENV endemic regions. Please also provide the time line, between which these samples were collected, and if there are any reports describing the serotypes circulating in the region at the same time window.

--I guess the samples were deidentified from the patient’s protected health information. This needs to be mentioned along with any approval taken from the institutional review board before working on human specimens.

--The distribution of all the serotypes is interesting. However, it would be very informative and useful if they are presented with a spatial or chronological distribution. That should answer questions like

  1. Did all 4 serotypes really infected the study population homogenously, or some areas got one serotype while other areas got another? You may not have this information if the patients’ residence data is not known. (can be presented as a map or table)
  2. During the range of the sample collections, were there waves of one strain followed by another with mixed infections at the transitions, or they were present homogenously all the time? (can be presented as a graph with time on x axis and case counts for each serotype on y axis).

Table 2 and 3 represent the most important findings of this study. Please add statistical conclusions like signs/symptoms were significantly/not different between the infections. And in discussion add information from literature if this study found the same or different.

Table 4 can be one statement with the range(s) of accession numbers, and the table become supplement, where they can be fully expanded and each row is an accession numbers along with the serotype. (if possible).

Figuere 1 is not necessary.

In the patients with multiple serotypes identified, what did the sequencing data say? If looks like you got more than one sequence from the 354 cases? This looks off: the submission list in table 4 shows almost a thousand sequences.  

Remove excess unnecessary words like “Without a doubt,” in Line 205.

Table 5: could you provide a phylogenetic tree of the sequences, which is the basis of this table. You could describe this table data if there are too many tables.

Round 2

Reviewer 1 Report

Thank you for your response. I am disappointed by the author's decision not to conduct additional experiments. I hope that future research will apply this suggenstion. However, the quality of the manuscript has improved to a certain level and is of considerable values. I believe that this manuscript deserves to be published in Viruses.